# ARVideo: Autoregressive Pretraining for Self-Supervised Video Representation Learning

## Abstract

This paper presents a new self-supervised video representation learning framework **ARVideo**, which *autoregressively* predict the next video token in a tailored sequence order. Two key designs are included. First, we organize autoregressive video tokens into clusters that span both *spatially* and *temporally*, thereby enabling a richer aggregation of contextual information compared to the standard spatial-only or temporal-only clusters. Second, we adopt a randomized spatiotemporal prediction order to facilitate learning from multi-dimensional data, addressing the limitations of a handcrafted spatial-first or temporal-first sequence order. Extensive experiments establish ARVideo as an effective paradigm for self-supervised video representation learning. For example, when trained with the ViT-B backbone, ARVideo competitively attains 81.2% on Kinetics-400 and 70.9% on Something-Something V2, which are on par with the strong benchmark set by VideoMAE. Importantly, ARVideo also demonstrates higher training efficiency, *i.e.*, it trains 14% faster and requires 58% less GPU memory compared to VideoMAE.

## 1 Introduction

The transformer architecture, as introduced in Vaswani *et al.* (Vaswani et al., 2017), has fundamentally transformed the field of natural language processing (NLP) through its ability to model long-range dependencies with minimal inductive bias. A crucial catalyst for its success lies in self-supervised learning of robust and transferable representations from large volumes of unlabeled data. Within this paradigm, masked language modeling (MLM) (Devlin et al., 2019) and autoregressive modeling (AR) (Radford et al., 2018; Brown et al., 2020; OpenAI, 2023) stand out as two leading approaches. Specifically, MLM masks random portions of input tokens and trains models to predict masked elements; whereas AR predicts subsequent words in a sequence based on all preceding words. These methods have propelled state-of-the-art performance in various NLP tasks.

In the video domain, however, the landscape is different. Previous studies have predominantly relied on supervised pretraining using image datasets, often overlooking the critical aspect of temporal dynamics (Liu et al., 2022b; Bertasius et al., 2021). Recently, there has been a shift towards leveraging NLP-inspired mask language modeling (Devlin et al., 2019) or image-inspired mask image modeling (He et al., 2022; Bao et al., 2022) to directly exploit unlabeled video datasets for pretraining. For instance, VideoMAE (Tong et al., 2022; Feichtenhofer et al., 2022) introduces mask autoencoder (He et al., 2022) for self-supervised video video representation learning; BEVT (Wang et al., 2022a) learns spatial representations from image data and joint-masked image and video modeling. Despite these advancements, autoregressive modeling—another powerful self-supervised learning approach in NLP—has yet to be extensively explored within the context of video data analysis.

Critically, applying autoregressive pretraining to video data entails the same principle of autoregressively predicting the next *element* in a sequential order based on its predecessors. In natural language, these elements—words—are clearly defined and inherently follow a chronological order. For images, elements could be conceptualized as pixels or patches arranged in a flattened sequence (Chen et al., 2020; El-Nouby et al., 2024; Ren et al., 2024). The further transition to video data, however, introduces additional complexity due to its inherently multidimensional nature (*i.e.*, including both spatial and temporal dimensions). This raises a crucial inquiry: *how should we define*

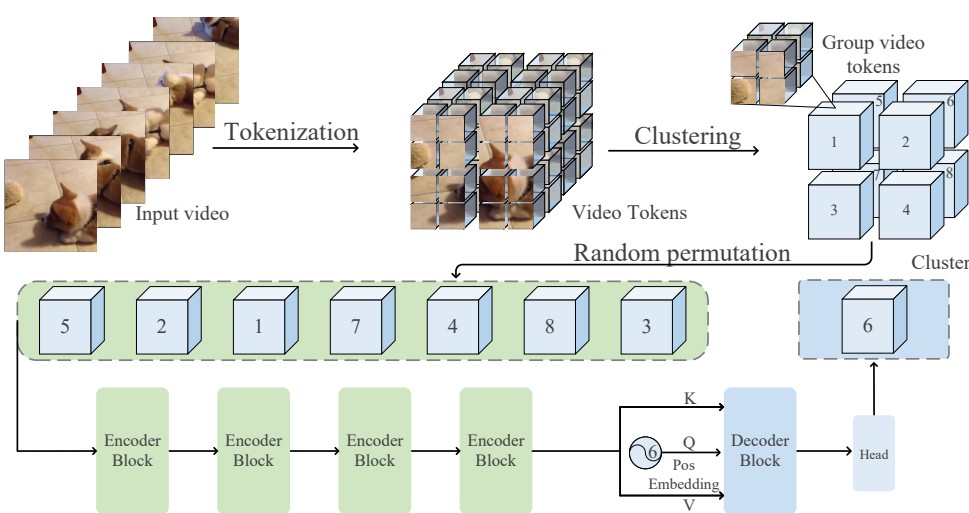

Figure 1: ARVideo autoregressive predicts spatiotemporal cluster from grouping tokens span spatial and temporal dimension.

*an autoregressive 'video element' and establish a visual sequence order for self-supervised video representation learning?*

We note traditional methods, such as converting video into a sequence of cubes (Tong et al., 2022; Bertasius et al., 2021; Wang et al., 2022a; Liu et al., 2022b) and subsequently linearly mapping these cubes into video tokens, generally reveal critical limitations in addressing this query. Specifically, the granularity of these video tokens often fails to encapsulate the rich semantics typically represented by words in text-based models—primarily because 1) these video tokens are too dimensionally limited, and 2) video inherently lacks a sequential order in its spatial dimensions, although it retains this feature in its temporal aspects.

To address these challenges, we hereby present **ARVideo**, a novel autoregressive-based video representation learning paradigm with two key designs (see Figure 1). Firstly, we redefine 'video elements' by grouping video tokens into spatiotemporal video clusters, differentiating from conventional single-dimensional strategies like spatial video clusters or temporal video clusters. This approach improves semantic representation by aggregating more contextually relevant multidimensional information. Secondly, we find that, compared to well-defined yet single-dimensional spatial-first or temporal-first sequence orders, a sequence order that randomly integrates both spatial and temporal dimensions empirically yields significantly stronger results. This suggests that effectively capturing the inherent multidimensionality of video data is crucial for autoregressive modeling. Extensive experiments establish our ARVideo as an effective paradigm for video representation learning. For example, while the autoregressive video representation learning baseline only attains 74.2% on Kinetics-400 and 66.4% on Something-Something V2, ARVideo significantly boosts the results to 81.2% (+7%) and 70.9% (+4.5%), respectively. Notably, these results not only match but, in some aspects, surpass the strong benchmark set by VideoMAE, particularly with respect to training efficiency—ARVideo achieves faster training speeds by 14% and reduces GPU memory consumption by 58%.

## 2 RELATED WORK

### 2.1 VIDEO REPRESENTATION LEARNING

Video representation learning has witnessed significant exploration, historically driven by supervised learning methods (Tran et al., 2018; Wang et al., 2019; Simonyan & Zisserman, 2014; Bertasius et al., 2021; Liu et al., 2022b) that pretrain backbone networks on labeled image or video data before

fine-tuning. However, such methods face challenges due to inherent discrepancy between image and video data, compounded by the scarcity of comprehensively labeled video datasets.

In the era of self-supervised learning, recent work have designed pre-tasks incorporating temporal information for self-supervised video representation learning (Xu et al., 2019; Benaim et al., 2020; Huang et al., 2021; Qian et al., 2021; Ranasinghe et al., 2022) and leveraging contrastive learning for effective visual representations (Qian et al., 2021; Kuang et al., 2021; Li et al., 2021; Diba et al., 2021; Han et al., 2020a;b). Additional, mask reconstruction-based methods inspired by masked language modeling (Devlin et al., 2019) are introduced into self-supervised image and video representation learning. For example, MAE (He et al., 2022) presents a scalable self-supervised learning method to reconstruct masked image patches while VideoMAE (Tong et al., 2022) extends this approach to video data and reconstructs masked spacetime patches. BEVT (Wang et al., 2022b) separates spatial learning from temporal dynamics, training on masked images initially before jointly on masked images and videos. Christoph *et al.* (Feichtenhofer et al., 2022) introduce an efficient video-based MAE extension with minimal biases and significant speedups. In contrast to prior works, our ARVideo proposes a new path for self-supervised video representation learning via autoregressive pretraining.

## 2.2 AUTOREGRESSIVE PRETRAINING

As a representative approach for autoregressive pretraining, Generative Pretrained Transformer (GPT) trains language models by autoregressively predicting the next word based on all preceding words in a sentence. Inspired by the success of autoregressive modeling in NLP, researchers start to apply autoregressive pretraining in computer vision. ImageGPT (Chen et al., 2020) learns effective image representations by training a Transformer to autoregressively predict image pixels without any prior knowledge of their 2D structure. SAIM (Qi et al., 2023) adopts an encoder to autoregressively learn contextual information like a standard vision transformer (ViT) and a decoder to predict the current content, mutually reinforcing each other's functions. RandSAC (Hua et al., 2022) arranges image tokens into segments for parallel intra-segment and sequential inter-segment autoregressive prediction. However, applying autoregressive pretraining on video data faces notable challenges due to the extra temporal dimension. ARVideo explores the design of autoregressive video elements and visual sequence orders for video representation learning.

## 3 METHOD

In this section, we first revisit GPT (Radford et al., 2018) and ImageGPT (Chen et al., 2020) to establish the foundation for the proposed ARVideo, as illustrated in Figure 1. We then analyze the inherent difference between image and video data, followed by the design of *elements* and the optimal prediction *order* as the key ingredients in ARVideo for autoregressive prediction with videos.

## 3.1 GENERATIVE PRETRAINED TRANSFORMER

We first outline the Generative Pretrained Transformer (GPT) framework. Consider an unlabeled language dataset $\mathcal{U}$ comprising sentences $[u^1, ..., u^N]$, where each sentence $u^j$ consists of words $u^j = \{u_1^j, ..., u_n^j\}$. GPT (Radford et al., 2018) autoregressively predicts the next word given all preceding words, minimizing the negative log-likelihood with model parameter $\theta$:

$$p(u^j) = -log \prod_{i=1}^{n} p(u_i^j | u_1^j, ..., u_{i-1}^j, \theta).$$

(1)

This modeling strategy has fundamentally changed the landscape of natural language processing, leading to the development of tremendously successful models like ChatGPT (Radford et al., 2018) and GPT-4 (OpenAI, 2023).

## 3.2 IMAGEGPT

Transitioning from natural language processing to image processing necessitates the design of image elements for autoregressive prediction. In ImageGPT, it treats individual pixels as elements. Specifically, given an image $x \in R^{H \times W \times C}$, ImageGPT flattens it into a 1D pixel sequence of length

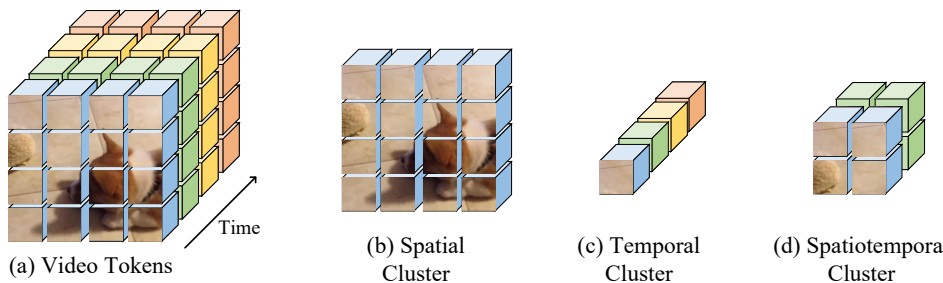

(a) Video Tokens        (b) Spatial Cluster        (c) Temporal Cluster        (d) Spatiotemporal Cluster

Figure 2: Comparison between video token and different cluster.

$N = H \times W$, and autoregressively predicts the next pixel given all preceding pixels:

$$p(x) = -log \prod_{i=1}^{N} p(x_i|x_1, ..., x_{i-1}, \theta) \qquad (2)$$

This approach incurs significant computational overhead due to the quadratic complexity of self-attention *w.r.t.* the input sequence length. ImageGPT thereby uses smaller image sizes (e.g., $32 \times 32$) in pretraining, yielding suboptimal performance. This limitation is pertinent in our development of ARVideo and becomes more pronounced due to the added complexity of video data.

## 3.3 ARVIDEO

Illustrated in Figure 1, ARVideo autoregressively pretrains on video data $x \in \mathcal{R}^{T \times H \times W \times C}$. Note that directly extending ImageGPT to videos faces significant challenges, primarily due to the added temporal dimension, which would significantly escalate computational demands, even with low-resolution videos like $4 \times 32 \times 32$. Moreover, pixels as autoregressive elements lack semantic richness compared to words in the language, further necessitating pixel grouping strategies to enhance representation learning. To better facilitate learning from multi-dimensional video data, we also explore prediction orders across spatial and temporal dimensions.

### 3.3.1 PIXEL GROUPING

**From Pixels to Video Tokens.** With patch embeddings in ViT, videos can be patchified into non-overlapping cubes (Tong et al., 2022; Bertasius et al., 2021; Wang et al., 2022a; Liu et al., 2022b) of size $P_T \times P_W \times P_H$. Then, each cube is transformed into a video token through a linear projection layer, resulting in $N = \frac{T}{P_T} \times \frac{H}{P_H} \times \frac{W}{P_W}$ video tokens. This tokenization significantly reduces operational elements, thus alleviating computational demands while ensuring that each video token encapsulates richer semantics compared to individual pixels. For example, as reported in Table 1, using video tokens as autoregressive elements for pretraining significantly outperforms approaches without tokenization by 3.3% while keeping pretraining resolution consistent with previous work (Tong et al., 2022; Wang et al., 2022a).

| Element | Resolution | Something- Something V2 |
|---|---|---|
| Pixel | $8 \times 14 \times 14$ | 60.7 |
| Token | $16 \times 224 \times 224$ | 64.0 |

Table 1: Grouping pixels into video tokens facilitates autoregressive pretraining on higher-resolution videos and improves performance by 3.3%.

This promising transition from pixels to video tokens introduces a compelling query: *Can further performance gains be realized by aggregating more tokens?* In pursuit of this, we examine three options: grouping video tokens into spatial, temporal, or spatiotemporal clusters. It is important

to note that within each cluster, video tokens are always fully attended to each other. This full-attention configuration helps to enable a more effective consolidation of semantic content within each autoregressive element.

**From Tokens to Spatial Clusters.** As shown in Figure 2(b), we strategically group spatially neighbored tokens—those sharing the same temporal positions but varying spatially—into spatial clusters. Following the patch embedding step, video tokens within the spatial domain $\frac{H}{P_H} \times \frac{W}{P_W}$ are grouped into one element, resulting in $\frac{T}{P_T}$ autoregressive elements. For example, a video of size $16 \times 224 \times 224$ with a cube embedding size of $2 \times 16 \times 16$ (Tong et al., 2022) here will be transformed into 8 autoregressive elements, with each element comprising $14 \times 14$ tokens.

**From Tokens to Temporal Clusters.** As illustrated in Figure 2(c), our method integrates temporal information by grouping tokens that are temporally adjacent into temporal clusters. Specifically, tokens within the temporal domain $\frac{T}{P_T}$ are grouped into one element, resulting in $\frac{H}{P_H} \times \frac{W}{P_W}$ autoregressive elements. For instance, a video of size $16 \times 224 \times 224$ with a cube embedding size of $2 \times 16 \times 16$ (Tong et al., 2022) here will transformed into $14 \times 14$ autoregressive elements, with each element comprising 8 tokens.

**From Tokens to Spatiotemporal Clusters.** Moving beyond the single-dimensional grouping strategies discussed above, we now consider the inherently multidimensional nature of video data by grouping neighboring $K_T \times K_H \times K_W$ tokens into spatiotemporal clusters with no overlaps, as illustrated in Figure 2(d). This strategy results in a total number of $\frac{T}{P_T K_T} \times \frac{H}{P_H K_H} \times \frac{W}{P_W K_W}$ clusters, with each containing both spatial and temporal information as an autoregressive element.

### 3.3.2 SPATIALTEMPORAL PREDICTION ORDER

For the spatiotemporal cluster, we further explore its prediction order. Specifically, this strategy is expected to yield $\frac{T}{P_T K_T}$ clusters at each spatial position, and $\frac{H}{P_H K_H} \times \frac{W}{P_W K_W}$ clusters at each temporal position.

**Pre-defined order.** We implement two systematic strategies: a spatial-first order and a temporal-first order. The spatial-first approach prioritizes autoregressive pretraining within the $\frac{H}{P_H K_H} \times \frac{W}{P_W K_W}$ spatiotemporal clusters along the spatial dimension, before transitioning to clusters in subsequent temporal positions. Conversely, the temporal-first approach prioritizes within the $\frac{T}{P_T K_T}$ spatiotemporal clusters along the temporal dimension, then proceeds to clusters in subsequent spatial positions.

**Random Rasteration.** Inspired by the random sentence permutation technique used in XLNet (Yang et al., 2019) for enhancing autoregressive pretraining, our random rasterization approach scrambles the order of clusters randomly during autoregressive pretraining. This method avoids the constraints of fixed sequential patterns, such as spatial-first or temporal-first, and allows ARVideo to adaptively model both long- and short-range spatial-temporal information. Such flexibility in autoregressive prediction orders not only captures the inherent multidimensionality of video data more effectively but also fosters a richer, more comprehensive video representation.

### 3.3.3 MODEL ARCHITECTURE

We adopt the ViT (Dosovitskiy et al., 2021; Tong et al., 2022) as the encoder. For the decoder, we take the Transformer decoder with cross attention but without self-attention. This design choice aims to simplify the decoding process, emphasizing interaction between the encoded inputs while reducing training costs. The query of the decoder is randomly initialized but includes position information to facilitate sequence generation. Our model utilizes a strategically designed attention mask as in previous work (Chen et al., 2020; Radford et al., 2018) to enable efficient autoregressive prediction in a parallel computation framework. When transferring to downstream tasks, we remove the decoder and only finetune the encoder.

| Method | Backbone | pretrain | Epoch | Frames | GFLOPs | Param | Top-1 |
|---|---|---|---|---|---|---|---|
| *Supervised pretraining* | | | | | | | |
| TANet (Liu et al., 2021) | ResNet152 | IN-1K | 100 | 16 | 242×4×3 | 59 | 79.3 |
| TDN$_{En}$ (Wang et al., 2021) | ResNet101 | IN-1K | 100 | 8+16 | 198×10×3 | 88 | 79.4 |
| TimeSformer (Bertasius et al., 2021) | ViT-B | IN-21K | 15 | 8 | 196×1×3 | 121 | 78.3 |
| Motionformer (Patrick et al., 2021) | ViT-B | IN-21K+K400 | 35 | 16 | 370×1×3 | 109 | 81.1 |
| Video Swin (Liu et al., 2022a) | Swin-B | IN-21K+K400 | 30 | 32 | 321×1×3 | 88 | 82.7 |
| *Mask video modeling* | | | | | | | |
| VIMPAC (Tan et al., 2021) | ViT-L | HowTo100M | 100 | 10 | N/A×10×3 | 307 | 77.4 |
| BEVT (Wang et al., 2022a) | Swin-B | K400 | 150 | 32 | 282×1×3 | 88 | 76.2 |
| VideoMAE (Tong et al., 2022) | ViT-B | K400 | 800 | 16 | 180×2×3 | 87 | 80.0 |
| VideoMAE (Tong et al., 2022) | ViT-B | K400 | 1600 | 16 | 180×2×3 | 87 | 81.5 |
| *Autoregressive pretraining* | | | | | | | |
| iGPT (Chen et al., 2020) | ViT-B | IN-1K | 300 | 16 | 180×2×3 | 87 | 61.2 |
| Randsac (Hua et al., 2022) | ViT-B | IN-1K | 1600 | 16 | 180×2×3 | 87 | 70.3 |
| TokenGPT† | ViT-B | IN-1K | 300 | 16 | 180×2×3 | 87 | 68.5 |
| TokenGPT† | ViT-B | K400 | 800 | 16 | 180×2×3 | 87 | 74.2 |
| ARVideo | ViT-B | K400 | 800 | 16 | 180×2×3 | 87 | 80.1 |
| ARVideo | ViT-B | K400 | 1600 | 16 | 180×2×3 | 87 | 81.2 |

Table 2: **Comparison with the state-of-the-art methods on Kinetics-400**. "Ex. labels ✗" means only *unlabelled* data is used during the pretraining phase. "N/A" indicates the numbers are not available for us. † indicates the implementation by us with the token replacing pixel in iGPT.

## 4 EXPERIMENT

### 4.1 DATASET AND IMPLEMENTATION DETAILS

We primarily evaluate ARVideo on Kinetics-400 (Kay et al., 2017) and Something-Something V2 (Goyal et al., 2017). Specifically, Kinetics-400 contains 400 classes and 260k videos of 10s, with 240k for training and 20k for validation; Something-Something V2 contains 174 classes with 169k videos for training and 25k for validation. While Kinetics-400 provides a broad spectrum of actions with minimal context, Something-Something V2 focuses more on the interaction of actions with objects.

For our experiments, we first pretrain a vanilla video Transformer (Tong et al., 2022) with ARVideo, and then fine-tune the pretrained model on the target action recognition datasets. Additionally, we assess the feature transferability on AvA v2.2 (Gu et al., 2018) and HMDB (Kuehne et al., 2011). AvA v2.2 is a human action localization dataset with 211k videos for training and 57k for validation; HMDB is a small video dataset with 3.5k videos for training and 1.5k videos for validation.

We follow the established protocol in prior work (Tong et al., 2022) to train our models. Instead of using negative log-likelihood as in GPT (Radford et al., 2018), we employ mean square error (MSE) loss to measure the discrepancy between the predicted and target cubes, as utilized in MAE (He et al., 2022). We randomly mask 80% tokens in each element to reduce the overall training costs; note that, unlike MAE or VideoMAE, we do not reconstruct those masked regions.

### 4.2 MAIN RESULTS

**Kinetics-400.** We pretrain the ViT-B backbone for both 800 and 1600 epochs on Kinetics-400, and report the corresponding results in Table 2. Notably, ARVideo attains 80.1% top-1 accuracy under 800 epochs and 81.2% top-1 accuracy under 1600 epochs, exhibiting significant improvements over previous autoregressive methods. Specifically, taking 1600-epoch-pretrained ARVideo for comparison, it outperforms iGPT, the baseline model, by a striking **+20.0%**, and Randsac, the previous state-of-the-art autoregressive model on images, by **+10.9%**. Additionally, compared to TokenGPT, which performs token-level autoregressive prediction, ARVideo showed advancements of **+12.7%** when TokenGPT was pretrained on an image dataset, and **+7.0%** when it was pretrained on the Kinetics-400 dataset.

Moreover, we note that ARVideo performs competitively against the strong benchmark—the mask video modeling method, VideoMAE. For example, the performance difference between ARVideo and VideoMAE is only 0.1% with 800 epochs of pretraining; this margin remains minimal at 0.3%

| Method | Backbone | Pretrain | Epoch | Frames | GFLOPs | Param | Top-1 |
|---|---|---|---|---|---|---|---|
| *Supervised pretraining* | | | | | | | |
| TEINet$_{En}$ (Liu et al., 2020) | ResNet50$_{\times 2}$ | IN-1K | 50 | 8+16 | 99×10×3 | 50 | 66.5 |
| TANet$_{En}$ (Liu et al., 2021) | ResNet50$_{\times 2}$ | IN-1K | 50 | 8+16 | 99×2×3 | 51 | 66.0 |
| TDN$_{En}$ (Wang et al., 2021) | ResNet101$_{\times 2}$ | IN-1K | 60 | 8+16 | 198×1×3 | 88 | 69.6 |
| SlowFast (Feichtenhofer et al., 2019) | ResNet101 | K400 | 196 | 8+32 | 106×1×3 | 53 | 63.1 |
| MViTv1 (Fan et al., 2021) | MViTv1-B | K400 | 100 | 64 | 455×1×3 | 37 | 67.7 |
| TimeSformer (Bertasius et al., 2021) | ViT-B | IN-21K | 15 | 8 | 196×1×3 | 121 | 59.5 |
| TimeSformer (Bertasius et al., 2021) | ViT-L | IN-21K | 15 | 64 | 5549×1×3 | 430 | 62.4 |
| ViViT FE (Arnab et al., 2021) | ViT-L | IN-21K+K400 | 35 | 32 | 995×4×3 | N/A | 65.9 |
| Motionformer (Patrick et al., 2021) | ViT-B | IN-21K+K400 | 35 | 16 | 370×1×3 | 109 | 66.5 |
| Video Swin (Liu et al., 2022a) | Swin-B | IN-21K+K400 | 30 | 32 | 321×1×3 | 88 | 69.6 |
| *Mask video modeling* | | | | | | | |
| VIMPAC (Tan et al., 2021) | ViT-L | HowTo100M | 100 | 10 | N/A×10×3 | 307 | 68.1 |
| BEVT (Wang et al., 2022a) | Swin-B | IN-1K+K400 | 150 | 32 | 321×1×3 | 88 | 70.6 |
| MaskFeat†312 (Wei et al., 2022) | MViT-L | K600 | 1600 | 40 | 2828×1×3 | 218 | 75.0 |
| VideoMAE (Tong et al., 2022) | ViT-B | SSv2 | 800 | 16 | 180×2×3 | 87 | 69.6 |
| VideoMAE (Tong et al., 2022) | ViT-B | SSv2 | 2400 | 16 | 180×2×3 | 87 | 70.8 |
| *Autoregressive pretraining* | | | | | | | |
| iGPT (Chen et al., 2020) | ViT-B | IN-1K | 300 | 16 | 180×2×3 | 87 | 54.3 |
| Randsac (Hua et al., 2022) | ViT-B | IN-1K | 1600 | 16 | 180×2×3 | 87 | 59.6 |
| TokenGPT† | ViT-B | IN-1K | 300 | 16 | 180×2×3 | 87 | 59.2 |
| TokenGPT† | ViT-B | SSv2 | 800 | 16 | 180×2×3 | 87 | 66.4 |
| ARVideo | ViT-B | SSv2 | 800 | 16 | 180×2×3 | 87 | 69.8 |
| ARVideo | ViT-B | SSv2 | 2400 | 16 | 180×2×3 | 87 | 70.9 |

Table 3: **Comparison with the state-of-the-art methods on Something-Something V2**. "Ex. labels ✗" means only *unlabelled* data is used during the pretraining phase. "N/A" indicates the numbers are not available for us. † indicates the implementation by us with the token replacing pixel in iGPT.

| Method | K400 → AVA v2.2 | K400 → HMDB |
|---|---|---|
| *Contrastive Learning* | | |
| MoCo | - | 67.9 |
| *Mask video modeling* | | |
| VideoMAE | 26.7 | 73.3 |
| *Autoregressive pretraining* | | |
| ARVideo | 26.9 | 74.1 |

Table 4: Comparison of model transferability. We first pretrain models on Kinetics-400, and then transfer them to AVA v2.2 and HMDB.

with 1600 epoch pretraining. These results validate the effectiveness of ARVideo as a pioneering autoregressive pretraining method in self-supervised video representation learning, equalling—and in some aspects surpassing—the performance of established mask modeling methods.

**Something-Something V2.** We pretrain the ViT-B backbone for 800 and 2400 epochs on the Something-Something V2 dataset. As reported in Table 3, ARVideo achieves top-1 accuracies of 69.8% and 70.9% for 800 and 2400 epochs, respectively, which are significantly stronger than prior autoregressive pretraining methods. For example, under 2400 epochs, ARVideo surpassed the baseline model iGPT by **+16.6%** and outperforms the best-performing image-based autoregressive method, Randsac, by **+11.3%**. It also surpassed TokenGPT pre-trained on image datasets by +11.7% and on the Something-Something V2 dataset by +4.5%. Additionally, when compared to the strong masked video modeling method VideoMAE, ARVideo also performs competitively in both 800 epochs of pretraining (*i.e.*, 0.2% accuracy difference) and 2400 epochs of pretraining (*i.e.*, 0.1% accuracy difference). Together with the observations in Kinetics-400, these results can establish ARVideo as a strong alternative to masked modeling approaches for video analysis.

**Transfer Learning.** To investigate the feature transferability of ARVideo, we transfer the model trained on Kinetics-400 to AvA v2.2 and HMDB. We can observe that ARVideo demonstrate strong transferability, achieving 26.9 mAP on AvA v2.2 and 74.1% Top-1 accuracy on HMDB—outperforming both VideoMAE and MoCo (see Table 4). For example, compared to VideoMAE, ARVideo shows (slight) improvements of 0.2% on AvA v2.2 and 0.8% on HMDB.

| Method | Encoder | | Decoder | | Training Time | GPU Memory |
|---|---|---|---|---|---|---|
| | Q | Key/Value | Q | Key/Value | | |
| VideoMAE | 160 | 160 | 1568 | 1568 | 145h | 41.3G |
| ARVideo | 300 | 300 | 1372 | 300 | **127h** (-12.4%) | **26.1G** (-36.8%) |

Table 5: The comparison of pretraining time and GPU memory.

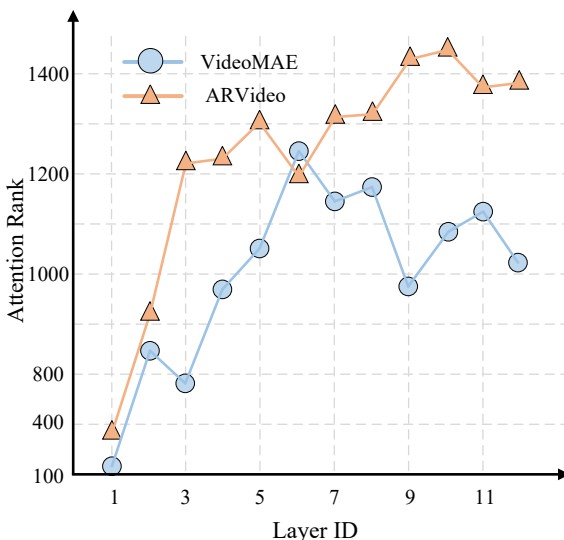

Figure 3: The attention rank comparison between VideoMAE and ARVideo

**Computation cost.** We report the training time and GPU memory usage in Table 5 (with ViT-B trained on Kinetics-400 for 800 epochs, using 8×A6000). Compared to VideoMAE, ARVideo presents significant reductions in both GPU memory usage and training time—ARVideo reduces training cost by 12.4% (from 145 hours to 127 hours) and GPU memory consumption by 36.8% (from 41.3G to 26.1G). This advantage stems from ARVideo's shorter sequence length as we drop the last cluster in the autoregressive modeling.

**Attention rank.** The self-attention mechanism computes attention scores for a given input sequence, forming what is known as the attention map. The rank of this matrix can serve as a measure of its ability to capture complex patterns in the data. Typically, high-rank attention matrices suggest a model that can capture a wide variety of patterns and relationships within the data, while low-rank matrices may suggest a model that does not well utilize its full capacity or operates on simpler data (Wang et al., 2020). Following this instruction, we plot the rank of the attention map in each layer of VideoMAE and our ARVideo in Figure 3. We can observe that, across nearly all layers except the $6_{th}$, ARVideo maintains higher attention ranks than VideoMAE, indicating a stronger representational ability of our model's self-attention layers.

**Visualization.** We provide some randomly selected visualization results in Figure 7. Note that this work aims to provide a new perspective to self-supervised video representation learning instead of video generation.

## 4.3  ABLATION STUDY

In this part, we ablate four factors—cluster shape, mask ratio, prediction order, and decoder design. Note that, unless otherwise specified, all ablations are conducted on the ViT-B backbone with 200 epochs of pretraining.

**Cluster shape.** We group neighboring and non-overlapped $K_T \times K_H \times K_W$ tokens into one cluster and analyze the effect of different cluster shapes. Three situations are considered: 1) $K_T = K_W = K_H = 1$, equivalent to the TokenGPT, which pertains autoregressively at the token/cube level; 2)

| case | $K_T$ | $K_H$ | $K_W$ | Something-Something V2 |
|---|---|---|---|---|
| Token/Cube | 1 | 1 | 1 | 64.0 |
| spatial cluster | 1 | $\frac{H}{P_H}$ | $\frac{W}{P_W}$ | 66.0 |
| spatial cluster | 1 | 7 | 7 | 66.2 |
| temporal cluster | $\frac{T}{P_T}$ | 1 | 1 | 65.2 |
| temporal cluster | 2 | 1 | 1 | 65.6 |
| spatiotemporal cluster | 4 | 7 | 7 | 65.5 |
| spatiotemporal cluster (ARVideo) | 2 | 7 | 7 | **66.8** |

Table 6: Ablation study on the cluster shape.

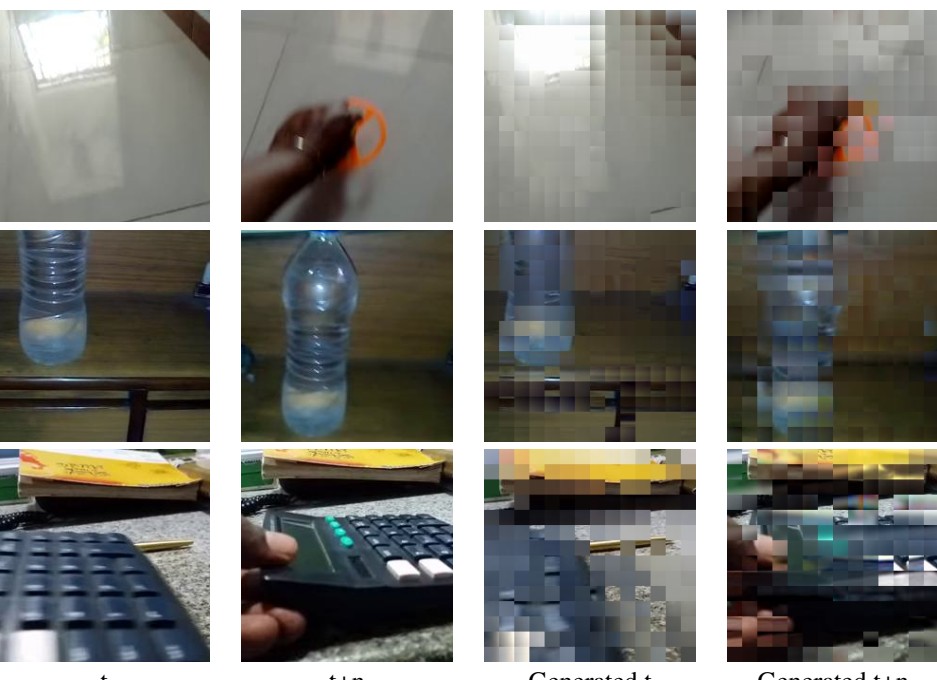

| t | t+n | Generated t | Generated t+n |

Table 7: The visualization results of GPT-Video.

$K_T = \frac{T}{P_T}, K_W = K_H = 1$, representing a temporal cluster; and 3) $K_T = 1, K_W = \frac{W}{P_W}, K_H = \frac{H}{P_H}$, representing a spatial cluster.

We report the results in Table 6. Firstly, we can observe that all clustered configurations significantly enhance performance over the TokenGPT baseline. For example, simply grouping tokens into spatial/temporal/spatiotemporal clusters yields 2.0%/2.2%/2.8% improvements, respectively. Then, when comparing different clusters, we note that our spatiotemporal cluster (ARVideo) with $K_T = 2, K_W = K_H = 7$ attains the best performance of 66.8%, outperforming the best-performed spatial cluster ($K_T = 1, K_W = K_H = 7$) by 0.8% and the best-performed temporal clusters ($K_T = 2, K_W = K_H = 1$) by 1.2%. However, it is interesting to note that, if a poorly designed spatiotemporal cluster ($K_T = 4, K_W = K_H = 7$) is used, the performance will drop to 65.5%.

**Prediction order.** In our evaluation of prediction order, which plays an important role in constructing the video sequence, we first check with the predefined spatial-first and temporal-first orders. As shown in Table 8, temporal-first order achieves 66.0% top-1 accuracy, which is 0.4% higher than spatial-first order. However, our randomized spatial-temporal prediction order, adept at learning both long- and short-range spatial-temporal dynamics, exhibits a superior performance of 66.8%, surpassing the predefined spatial-first approach by 1.2% and the temporal-first approach by 0.8%.

**Mask Ratio.** To reduce the temporal redundancy, ARVideo randomly mask a portion of tokens as in Flip (Li et al., 2023), MAE (He et al., 2022) and VideoMAE (Tong et al., 2022). We hereby check

| Order | SSv2 |
|---|---|
| Spatial-First | 65.6 |
| Temporal-First | 66.0 |
| Spatial-temporal random | **66.8** |

Table 8: Ablation study on the prediction order.

| Mask Ratio | SSv2 |
|---|---|
| 75% | 66.0 |
| 80% | 66.8 |
| 90% | 65.6 |
| 95% | 64.8 |

Table 9: Ablation study on the mask ratio from 75% to 95%.

| Method | Decoder | | Something-Something V2 |
|---|---|---|---|
| | Self-Atten | Cross-Atten | |
| ARVideo | | ✓ | 66.8 |
| ARVideo | ✓ | ✓ | 66.6 |

Table 10: Ablation study on the decoder architecture.

| Decoder Width | Decoder Depth | Something-Something V2 |
|---|---|---|
| 384 | 4 | 66.0 |
| 512 | 4 | **66.8** |
| 768 | 4 | 66.8 |
| 512 | 2 | 66.2 |
| 512 | 4 | **66.8** |
| 512 | 8 | 66.6 |

Table 11: Ablation study on the decoder depth and width.

how the masking ratio affects the overall performance. As shown in Table 9, our study starts from a mask ratio of 75% (*i.e.*, same as the MAE's setup), which achieves 66.0% top-1 accuracy. Increasing the mask ratio to 80% boosted the top-1 accuracy to 66.8%, while further increases to 90% and 95% lower the top-1 accuracies by 1.2% and 2.0%, respectively. We stress that, although ARVideo used a lower mask ratio than VideoMAE, it still enjoys faster training speeds and reduced GPU load (see Section 4.2 and Table 5).

**Decoder Architecture.** We hereby explore the effects of different decoder architectures. As reported in Table 10, whether or not having self-attention in the decoder has little effect on performance (*i.e.*, 66.6% *vs*. 66.8%), but excluding self-attention significantly reduces computational costs. Therefore, we take the decoder without self-attention by default in ARVideo.

**Decoder Width and Depth.** Lastly, we systematically ablate two critical aspects in designing decoders: its *width* and *depth*. We start with a four-layer decoder and follow the default setup in VideoMAE. As presented in Table 11, increasing the decoder width shows performance improvement from 66.0% at a width of 384 to 66.8% at a width of 512. Further width increase makes the performance plateau. Meanwhile, in terms of depth, deviations from the four-layer standard negatively impacted performance: *e.g.*, increasing to eight layers decreased performance by 0.2%, while reducing to two layers dropped performance by 0.6% (see the last three rows in Table 11).

## 5 CONCLUSION

In this paper, we introduce ARVideo for self-supervised video representation learning, inspired by the autoregressive principles of GPT in natural language processing. Diverging from conventional methods, our approach innovatively uses video token clusters as the element for autoregressive prediction, significantly reducing computational demands while still managing to capture essential spatial-temporal dynamics. This advancement improves the efficiency of video data processing and sets a new paradigm for self-supervised video representation learning. The promising results obtained from ARVideo underscore its potential and advocate for further exploration and development of autoregressive pretraining methods within the video domain.

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
