# OpenReview forum: "ARVideo: Autoregressive Pretraining for Self-Supervised Video Representation Learning"
_ICLR.cc/2025/Conference — ICLR 2025 Conference Withdrawn Submission_

### Official Review · Reviewer_7dvJ · 2024-10-28

**Soundness:** 3
**Presentation:** 3
**Contribution:** 3
**Rating:** 3
**Confidence:** 4

**Summary:**

This paper proposes a video representation learning framework using autoregressive (AR) modeling. The authors introduce two key innovations: grouping adjacent spatiotemporal tokens into a single unit and adopting a randomized prediction order for more effective learning. Their model achieves competitive performance on standard video classification benchmarks, demonstrating both accuracy and training efficiency.

**Strengths:**

- The paper is well-written and easy to follow.
- The authors conduct extensive ablation studies, effectively demonstrating the benefits of their approach.

**Weaknesses:**

- The paper lacks a clear explanation of the rationale behind the performance gains from grouping neighboring tokens (Table 6).
- Although the method includes masking 80% of tokens within each unit, the paper underemphasizes its reliance on mask-based modeling.
- The reported improvements in accuracy and efficiency are not significant compared to VideoMAE. Furthermore, the authors do not compare their method with VideoMAE-v2, a more recent and stronger baseline.

**Questions:**

- Can the process of grouping adjacent spatial-temporal tokens into a single unit truly be considered "clustering"?
- What exactly drives the performance gains from grouping tokens together? Is there an intuitive explanation?

---

### Official Review · Reviewer_qzna · 2024-10-31

**Soundness:** 2
**Presentation:** 3
**Contribution:** 1
**Rating:** 3
**Confidence:** 4

**Summary:**

This paper introduces ARVideo, a self-supervised video representation learning framework that autoregressively predicts video tokens. It uses spatiotemporal clusters for richer context and a randomized prediction order to enhance learning. ARVideo achieves competitive results on benchmarks like K400 and SSv2 while being more training efficient compared to VideoMAE.

**Strengths:**

1. This paper is well-organized, with a clear introduction, method, and experimental results sections, making it easy to follow.
2. The experimental results provide evidence to support the claims about ARVideo's performance and efficiency.

**Weaknesses:**

1. This paper should provide stronger justification for choosing autoregressive techniques over other self-supervised methods like contrastive learning and mask modeling.
2. Although this paper uses autoregressive modeling for self-supervised video representation learning, it offers incremental novelty by incorporating masking techniques in an autoregressive form. This approach refines existing mask modeling methods, using AR to predict masked tokens, which enhances contextual learning but may not significantly advance the methodologies in the field.
3. The effect of auto-regressive modeling is unclear, the provided experimental results cannot demonstrate the advantage of AR compared to mask modeling methods.
4. Even compared to the methods in Tab.1 and Tab.3, e.g., VideoMAE, the performance of the proposed method is marginally improved. Moreover, more recent methods are missing for comparison and discussion, e.g., ST-MAE [1], OmniMAE [2], MVD [3], MME [4].

[1] Feichtenhofer C, Li Y, He K. Masked autoencoders as spatiotemporal learners. NeurIPS2022.

[2] Girdhar R, El-Nouby A, Singh M, et al. Omnimae: Single model masked pretraining on images and videos. CVPR2023.

[3] Wang R, Chen D, Wu Z, et al. Masked video distillation: Rethinking masked feature modeling for self-supervised video representation learning. CVPR2023

[4] Sun X, Chen P, Chen L, et al. Masked motion encoding for self-supervised video representation learning. CVPR2023

**Questions:**

1. How to finetune the pre-trained ARvideo model for downstream tasks? Would it also be in an auto-regressive manner? How is the fine-tuning efficiency?
2. What are the results if we only apply the proposed cluster and masking strategy to mask modeling methods such as VideoVAE?

---

### Official Review · Reviewer_gcrn · 2024-11-02

**Soundness:** 3
**Presentation:** 2
**Contribution:** 2
**Rating:** 5
**Confidence:** 4

**Summary:**

The authors propose a new self-supervised video representation learning framework referred to as ARVideo, which autoregressively predicts the next video token in tailored sequence order.

First, the ARVideo organizes autoregressive video tokens into spatial and temporal clusters to obtain richer contextual information than the conventional clusters.

Second, ARVideo adopts a randomized spatiotemporal prediction order to facilitate learning from multidimensional data and address the limitations of the conventional handcrafted spatial-first or temporal-first sequence order.

Extensive experiments demonstrate that ARVideo is a practical paradigm for self-supervised video representation learning with higher training efficiency.

**Strengths:**

(+) The author introduces a new autoregressive pretraining framework to obtain richer contextual video representations in a self-supervised manner.

**Weaknesses:**

(-) We need a deeper analysis of the higher attention rank scores for ARVideo shown in Fig. 3, along with details on the benchmark dataset and hyperparameters used for ARVideo. Specifically, which components of ARVideo contributed to its higher rank score?

(-) Additionally, we require a more detailed interpretation of spatial-temporal randoms from an autoregressive training perspective.

(-) On a minor note, the figure captions lack sufficient detail. For example, in Fig. 1, while the head and target block 6 are clear visually, further explanation is needed for clarity. The same applies to Fig. 7.

**Questions:**

Please see the above weakness.

---

### Official Review · Reviewer_U2E2 · 2024-11-03

**Soundness:** 2
**Presentation:** 3
**Contribution:** 2
**Rating:** 3
**Confidence:** 4

**Summary:**

This paper presents a new self-supervised learning method for video representation learning which consists in training an auto-regressive transformer on top of clustered features from input video data. The clustering operation consists in aggregating video tokens locally and an auto-encoder is trained to predict auto-regressively the tokens in a randomized order. The method is evaluated on video classification tasks on K400 and SSv2, and is compared to VideoMAE in terms of performance, and computational efficiency.

**Strengths:**

- The proposed method exhibits an improved training time and memory usage compared to VideoMAE, while achieving the same level of performance on downstream tasks. A clear comparison between the methods is presented.

**Weaknesses:**

- I don’t really get what is interesting with the randomized order. It is good for learning invariant features, and for global tasks such as classification, but this is not really a novelty, the video SSL community is now shifting toward the more interesting idea of making causal prediction, which in the long-term is way more interesting. Overall it also seems that the prediction setup with the proposed method does not impact the performance much in the ablations, maybe this is also related to using the fine-tuning protocol, as I discuss below.

- All the comparisons with concurrent methods and the ablation are done with the fine-tuning protocol. This protocol is still by the video self-supervised learning community because most methods perform badly on the more interesting frozen evaluation protocol. However it is very uninformative as most of the time the performance comes from the tuning of the fine-tuning pipeline. Recent work such as V-JEPA has shown that video SSL models can also perform well on frozen tasks and the community should care more about these metrics that are the gold standard in SSL from images.

- Using “clustering” is very misleading as there is no active mechanism to cluster the tokens, but rather they are aggregated with a fixed rule depending on location. It is also not clear in the paper, what is the aggregation operation, is it average pooling or max pooling or something else ?

- Table 2 and Table 3 show that the new method is not better than VideoMAE, and it is not surprising as predicting autoregressively in random order does not have more expressivity in terms of learning compared to predicting everything randomly at once as in VideoMAE.

- Table 5 is extremely unclear, what are Q and Key/Value ?

- Visualizations of Table 7 are not very interesting and do not bring anything while taking a lot of space, maybe move them in appendix.

**Questions:**

- In Figure 3, what is the rank metric used in practice ? Does it really correlate with the performance ? For example when training a probe at a particular layer ?

---

### Official Review · Reviewer_zZpe · 2024-11-04

**Soundness:** 2
**Presentation:** 2
**Contribution:** 2
**Rating:** 3
**Confidence:** 4

**Summary:**

They group video tokens into spatiotemporal video clusters for pretraining, then randomly reorder the clusters in spacial and temporal directions to train the model.

**Strengths:**

The proposed method is aware of the advantages of using spatiotemporal data and randomness for pretraining, which yields better results on Kinetics dataset.

**Weaknesses:**

1.	The clustering of video tokens just group them with their neighbors in different directions. Many research works have explored spatiotemporal methods, and video tokens themselves are already spatiotemporal.
2.	Using a random order for better results is simply an empirical finding. Does the benefit of Random Rasteration come from data augmentation? Please explain theoretically.
3.	Overall, I think the methods are straightforward and the contribution is not enough.

**Questions:**

What is the difference between this clustering method and using a larger video token?

---

### Note · Authors · 2024-11-16

I have read and agree with the venue's withdrawal policy on behalf of myself and my co-authors.